# Hearing the Moment with MetaEcho!
# From Physical to Virtual in Synchronized Sound Recording

## ABSTRACT

In film education, high expenses and limited space significantly challenge teaching synchronized sound recording (SSR). Traditional methods, which emphasize theory with limited practical experience, often fail to bridge the gap between theoretical understanding and practical application. As such, we introduce MetaEcho, an educational virtual reality leveraging the presence theory for teaching SSR. MetaEcho provides realistic simulations of various recording equipment and facilitates communication between learners and instructors, offering an immersive learning experience that closely mirrors actual practices. An evaluation with 24 students demonstrated that MetaEcho surpasses the traditional method in presence, collaboration, usability, realism, comprehensibility, and creativity. Three experts also commented on the benefits of MetaEcho and the opportunities for promoting SSR education in the metaverse era.

## CCS CONCEPTS

• **Human-centered computing** → **Virtual reality**; **User studies**.

## KEYWORDS

Sync Sound, Presence, Virtual Reality, Arts Education

### ACM Reference Format:
Anonymous Author(s). 2024. Hearing the Moment with MetaEcho! From Physical to Virtual in Synchronized Sound Recording. In ,. ACM, New York, NY, USA, 9 pages.

## 1 INTRODUCTION

The conventional approach of instructing *synchronized sound recording* [5] involves using Microsoft PowerPoint presentations and relying on the instructor's expertise. In the usual classroom setting, learners can only engage in rudimentary film sound recording tasks inside a restricted yet degraded setting [21, 23]. It is worthwhile highlighting that sound is invisible in the physical world. Thus, the operations of documenting sound become abstract, and on-site training in film studios is essential for students studying *synchronized sound recording* (SSR). The on-site training allows students to acquire both technical and artistic skills, as well as the ability to apply abstract acoustic theory in real-world situations. In addition, the critical constraints of the high cost of industrial-grade equipment and unavailable professional venues, such as professional-grade studio facilities, deteriorate the learning experiences of SSR.

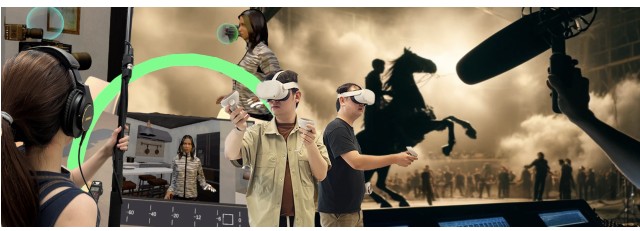

**Figure 1: MetaEcho's SSR education.**

The verbal discussion is considered a less effective approach, as the hands-on exercise considers the understanding of sound attributes and the techniques of operating the microphone to acquire audio sources and document sound effects. Nonetheless, most art schools employed sole verbal discussion that impedes students' capacity to proficiently acquire the technical skills of SSR, leading to a notable decline in instructional efficacy and practicality [3, 28]. An obvious example is that students and teachers can only rely on verbal descriptions and in-class discussions. With static visual aids, teachers demonstrate classic films as case studies. Such a teaching mode hinders the learning effects due to a lack of hands-on experience, even though the related knowledge and concepts are reviewed repeatedly in the classroom.

Due to the constraints of current instructional techniques, it is crucial to investigate alternative educational technologies for learning SSR that meet industrial standards. Therefore, we design and implement a virtual studio named MetaEcho (Figure 1). It is a virtual reality (VR) platform that serves as the first effort to assist learning and teaching for synchronized sound recording. The platform offers learners a wide range of educational materials via its immersive experience and task design that covers the fundamentals of SSR and cinema techniques. MetaEcho's instructional packages are specifically created to improve the learners' perception of spatial effects in sound design, involvement, and genuineness by visualizing sound, ultimately enhancing student learning performances. Figure 2 illustrates how our system creates an engaging learning environment by overcoming the constraints of physical equipment in real-time recording settings. Our virtual environment aims to enhance students' comprehension of *synchronized sound recording* through visualisation and facilitate the teaching and learning of contemporary film recording. The MetaEcho system consists of two major features. In MetaEcho's virtual studio, instruction and training for film recording begin. Students can immerse themselves in detailed simulations of film production sets and a variety of recording equipment. Using a visual representation of sound, students can capture and document the source of the audio precisely. Furthermore, the system automatically provides instant feedback to evaluate the accuracy of students' recording operations, thus enhancing the instructional value of the learning film recording process. Twenty-four students were recruited to assess the learning

outcomes and operation proficiency between the MetaEcho system and the traditional lecture using PowerPoint slides.

Specifically, our contributions are as follows. First, MetaEcho is an immersive platform specifically for teaching SSR under the intersection of film production and education. It offers comprehensive virtual tools and appropriate feedback for improving students' technical skills, applying filming theories, and creative thinking. Second, the teaching mode of MetaEcho employs acoustic simulation that offers a sense of presence to enhance the learning process. In a virtual studio that closely resembles real-life scenarios, students can freely practice with virtual entities, which effectively lowers the expenses associated with learning while also increasing its accessibility and flexibility. Third, our assessment, which included 24 students and three experts, revealed that the students appreciated MetaEcho's effectiveness in applying acoustic theory to real-life scenarios. Meanwhile, the experts validated the benefits of MetaEcho in augmenting learning effectiveness and fostering a better standard of SSR education.

## 2 RELATED WORKS

***Synchronized Sound Recording Teaching***. Since the transition from silent to sound films, advancements in sound recording technology have transformed film production and profoundly affected the viewing experience [9, 12]. The advent of digital technology has evolved film sound recording from mechanical to high-definition digital methods [31, 37, 49]. In the teaching process of synchronized sound recording (SSR), the current approach primarily focuses on theoretical instruction combined with case studies and some practical operations [21, 23], aiming to deepen students' understanding of SSR and to cultivate their ability to apply theoretical knowledge in actual production. However, most institutions face high costs from training equipment and confined space. This results in an educational approach prioritising theory over practice, leading to relatively few opportunities for students to engage in hands-on classroom practice. Therefore, MetaEcho serves as a first effort to alleviate the aforementioned issues.

***VR for art education***. Virtual reality (VR) instruction is extensively used throughout diverse sectors in contemporary times [8, 13, 17, 41]. Kavanagh et al. and Lee et al. [18, 22] found that VR benefits students' motivation and other cognitive capacities. Since 2000, VR has been first introduced into the field of art education[4], and now showcases increasing examples [6, 15, 47]. VR teaching offers distinct benefits, resolving the issues of traditional learning with PowerPoint (PPT) presentations, including ignoring student engagement, which hampers effective communication [19, 50]. VR, in comparison, offers an immersive learning environment that not only encourages active student engagement but also improves interaction and instructional adaptability in the classroom [27, 33, 34]. In applying VR to a 3D drawing education, the VR-enabled learning process is easier and more efficient due to its intuitive interface and immersive experience [6]. Furthermore, VR instruction enables students to independently explore and learn, promoting enhanced assimilation and understanding of knowledge [10, 26, 32]. These results emphasise the capacity of VR environments to revolutionise teaching methods and showcase its distinct advantages in tackling learning obstacles, e.g., introducing abstract concepts of arts education. One latest work [47] demonstrates clue of integrating film's

lighting equipment into VR settings, enabling students to have a very realistic learning experience of cinematography lighting. However, VR is rarely used for delivering learning and training content of *synchronized sound recording* [5]. Our work, therefore, aims to explore VR usage to improve students' interactive and participatory learning experiences in *synchronized sound recording*. Additionally, a simulated environment can potentially boost the learning effectiveness of operating a shotgun mic and further improve students' comprehension of sound attributes. A virtual studio for practising sound recording techniques becomes necessary [30, 35]. Meanwhile, our study reinforces the findings in prior work, showing that VR outperforms PPTs, in terms of student engagement, interaction, and understanding of SSR.

***Sense of Presence in VR Teaching***. Schubert et al. [43] analysed the sense of presence by categorising it into three primary dimensions: spatial presence, Involved in, and authenticity. Their framework provides a concise overview of the fundamental aspects influencing users' perception and experience of virtual reality settings. The sense of presence, featured by immersiveness, has been extensively used in distance education, internet conferencing, psychotherapy, and virtual reality [7, 14, 29, 38]. This theory examines the potential of technology to increase users' perception and engagement through an immersive experience, therebyinfluencing their learning, communication, and behavioural performance. In education, a greater presence may foster heightened participation in the learning process and learning results [24]. For instance, Sommer et al. [45] showcased VR medical education that simulate surgical operations, resulting in enhanced skills, increased student engagement, and other learning outcomes. Our work discovers a niche yet unexplored area by leveraging the sense of presence and building a virtual studio named MetaEcho that assists the knowledge acquisition of *synchronized sound recording*.

## 3 SYSTEM DESIGN OF METAECHO

Hands-on experience is vital for teaching and learning film's synchronized sound recording (SSR). Both academia and industry strive to find effective ways to use emerging technologies that enhance the teaching and learning of synchronized sound recording. This section explains the design of the MetaEcho system, characterized by key concepts such as a sense of presence, user interaction, and teaching designs.

### 3.1 Sense of Presence

In various virtual worlds, experiencing the sense of presence is of utmost importance, connecting to the effects of learning experiences and outcomes [44]. Accordingly, MetaEcho contains an exceptionally engaging instructional system. This system presents a virtual studio to instruct the synchronized sound recording of films. The spatial presence of the system allows students to immerse themselves in the learning experience, creating a sense of being physically present [20]. In addition, spatial presence, also known as spatial immersion, refers to the capacity of students to experience a state of complete involvement in a virtual environment. Therefore, within the current prototype of MetaEcho, we use the design principle of *Réalisme* [15] to provide students with an exceptional spatial immersive experience [40]. Our studio replicates a genuine

film industry production setting, including model construction, material mapping, and studio lighting, all meticulously designed to accurately resemble the actual environment. Thus, in the current prototype, we meticulously replicated the kitchen and dining room sequences from the film "It's Complicated" (2009). We ensured that the size, materials, and reflectivity of the virtual items and scenes closely resembled those in real life. More importantly, we replicate real-life equipment as much as possible, for the sake of enhancing the effectiveness of learning and teaching.

This particular scene is regarded as a common scenario in which characters make a series of dialogues in most movies. Additionally, it offers a diverse range of ambient sounds, such as those coming from outside the window and the TV in a restaurant. These elements provide valuable material for teaching filmmaking and contemporary recording techniques. Thus, MetaEcho owns a diverse range of recording equipment and adaptable material to enhance the student's interest. As such, authenticity is achieved by learning in the virtual studio that closely mirrors the real environment, including the participant's motions and the actual operation of the equipment.

To better simulate the actual process of film creation, the architecture of MetaEcho includes the simulation of the film camera and the operation of the *synchronized sound recording*. By implementing an automated motion system, the avatar may traverse a predetermined trajectory, replicating the actor's actual motions. Simultaneously, the camera's dynamic tracking guarantees coordination with the actor, i.e., through a representative avatar. During the simultaneous recording operation, the students use a monitor to view the live image from the camera. This allows them to adjust the shotgun mic's position to ensure high-quality recording in which the shotgun mic does not obstruct the camera view. This process simulates a real simultaneous recording scenario. MetaEcho showcases the practicality of collaborative learning via the simultaneous recording of a film by several users in a virtual studio, which is adaptable to various simultaneous recording configurations.

## 3.2 MetaEcho Interaction Pipeline

We summarize the interaction pipeline of MetaEcho into three modules, accompanied by a demonstration video[1].

**Spatial Presence**: The Spatial Presence Module immerses users in the virtual environment, aiming to elicit equivalent responses. The layout design of the virtual studio allows students to operate SSR equipment, providing a realistic experience. Based on Unity's advanced rendering functions, the scenes of hands-on exercises are highly consistent with the real world regarding environmental layout, audio source effects, and operational experience.

**Involvement**: The Involvement module aids students in focusing on synchronized sound recording studies through a user-friendly interface. Upon logging in, teachers can create a room, and students can join accordingly. The class begins once the teacher clicks "Ready," as illustrated in Figure 2 (1a – 1c). Figure 2 (2) depicts a film studio in our system, using real scenes as a blueprint. As illustrated in Figure 2 (3), the shotgun mic is visually represented by a white circular ring to indicate its sound capture range, and students can adjust the angle of the shotgun mic within the system using the A/B buttons on the right-hand controller to better aim at

the audio source. The system introduces two important concepts in synchronized sound recording: peak level and average level (Figure 2 (4)). Figure 2 (5) shows three common types of sounds with visual cues for *synchronized sound recording* practice: the human voice (green), exterior window effects (ambient sound effects) (yellow), and music (blue).

Figure 2 (6) depicts the virtual panel in MetaEcho, in which students can adjust the RMS level of the synchronized sound recording from -60dB to 0dB by moving the joystick on the right-hand controller forwards and backwards. Accordingly, the real-time sound feedback in the VR environment changes. Moreover, the system features an automatic detection function for synchronized sound recording to help students and teachers better assess the training effect of sound recording. Next, when the student's shotgun mic is not correctly positioned to an audio source, a red circle appears to signify the mistakes made by the student (Figure 2 (7a)). Figure 2 (7b) illustrates an orange circle highlighting the poor recording quality even if the shotgun mic is close to the correct audio source. When the shotgun mic aims at the audio source accurately, a green circle visualizes the correctness (Figure 2 (7c)). Finally, teachers can monitor the shooting perspectives through three views, while students can practice synchronized sound recording from different focal lengths and lens perspectives, as shown in Figure 2 (8).

**Realness**: The Realness module delivers an authentic experience in movement and scenes by providing virtual learning environments closely resembling the physical world [46], along with real-time synchronized sound recording capabilities. Authenticity is emphasized in our virtual education scenario of synchronized sound recording, ensuring users' actions mirror real-life operations. For example, users simulate boom operation by lifting controllers, integrating actions with a virtual film camera that mimics real shooting practices, as shown in Figure 2 (6). The system accommodates either spatial or confined environments by offering two action modes: (a) teleportation in a cramped space using a controller button, and (b) free movement in spacious settings for safety. These approaches simplify the learning process with a well-organized environment, ensuring an efficient transition for users.

## 3.3 MetaEcho Teaching Design

In general, teachers and students use the MetaEcho system to engage in communication and collaboration. This system allows users to learn either individually or in teams using a virtual studio. It is specifically designed to help students improve their skills in recording SSR for films. This indicates that the teaching approach is one-to-many, meaning that a single teacher is accountable for training several pupils. Meanwhile, our system is able to facilitate the instruction content consistently for all students. Thus, the syllabus was primarily created to support a teaching model where one instructor teaches multiple students simultaneously.

One of the authors in this article has undergone formal film education in synchronized sound recording and has three years of professional experience in the field. We searched online for publicly accessible course syllabi on synchronized sound recording to locate a viable model for a synchronized sound recording course [16, 21, 37]. Following extensive discussion between the authors and specialists, we have reached a consensus on a synchronized sound recording syllabus specifically designed for conventional

---

[1]https://anonymous.4open.science/r/MetaEcho-475B/README.md

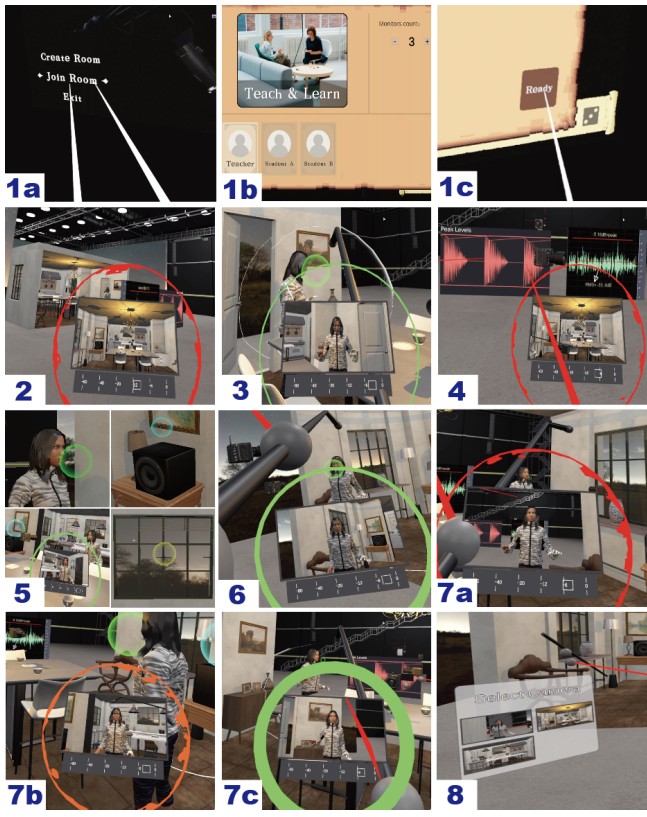

**Figure 2: MetaEcho offers educational experiences including: (1a)-(1c) Login and role selection; (2) Training scenarios; (3) Angle adjustment and collection range of shotgun mic; (4) Concepts of peak and average levels; (5) Demonstration of three types of SSR sounds; (6) Adjustment of RMS levels of SSR (range from -60dB to 0dB), practicing SSR and photography coordination, with real-time feedback in a VR environment; (7a)-(7c) Indicating the accuracy of the shotgun mic's alignment with the audio source using red, orange, or green circles; (8) Providing simulation training with three different focal lengths.**

teaching using PowerPoint presentations (PPT). The curriculum for cinematic simultaneous recording was modified to accommodate both conventional teaching settings (utilising traditional classroom settings with audio-visual instruction using PPT) and virtual reality teaching (on the MetaEcho platform). These modifications meant that the curriculum could fully use the functionalities of the MetaEcho platform without altering the primary content or overall duration of the course.

Before the course begins, the teacher allocates 5-10 minutes to familiarise the students with the virtual reality controllers and their functions. During this time, the students can navigate the MetaEcho environment and play with the virtual instances in MetaEcho. It is important to note that this practice session is not included in the overall course duration. At the beginning of the lecture, we adhere to a particular SSR course curriculum equivalent to a single 50-minute session. This lecture is divided into four parts:

*Part 1 (10 minutes) An overview of Simultaneous Recording Equipment*: The MetaEcho platform provides a comprehensive range of recording equipment, such as a shotgun mic, boom pole, recorder, and monitor headphones. The instructor familiarises users with the functions and features of these different equipment in the virtual environment. Additionally, the instructor discusses the applicable scenarios, considerations, justifications, and pros and cons of using the equipment for the *synchronized sound recording*. Furthermore, the teacher demonstrates these gadgets inside the virtual setting to students to illustrate the correct way of holding the boom pole for filming, as well as the effective utilisation of the shotgun mic for locating the audio source.

*Part 2 (10 minutes) An overview of three distinct sound categories under simultaneous recording conditions*: acoustic voices, sound effects, and music. The teacher uses the MetaEcho platform to showcase the creation of various sounds and their distinctiveness to students. This also enables students to see the recording process in real-time. Subsequently, the teacher presents two fundamental principles, as follows. The first concept is named **Peak Level** – The notion of peak level is crucial in the context of the *synchronized sound recording* in film as it ensures that the captured sound closely resembles the original without any distortion. This meticulous adjustment of the volume control is essential for the audience's auditory satisfaction. Regulating the maximum levels results in a more harmonious and lifelike auditory impact. The second concept refers to **RMS Level** – The RMS Level is a significant metric that quantifies the average power level of a soundtrack, specifically emphasising the overall loudness and consistency. The RMS Level, in contrast to peak levels, guarantees a steady and authentic soundtrack across different scenes, preventing listener fatigue and delivering a well-balanced and lifelike audio experience. By regulating the average amplitude, the soundtrack's volume may be modified without compromising its distinctness, resulting in a more pleasant and authentic auditory encounter. In Figure 2 (6), the instructor adjusts the RMS levels to -6dB, -12dB, and -18dB while recording vocals and sound effects. This demonstrates to students how RMS levels influence sound quality and perception, highlighting the importance of volume management in creating immersive audio experiences.

*Part 3 (20 minutes) Audio Sources and Recording Techniques*: In Part 2, we provide a comprehensive overview of the vocal characteristics and simultaneous recording techniques for three distinct types of sounds. These include three types of vocals, namely dialogues, voice-overs, and various character voices. Additionally, the students discuss three types of sound effects, such as ambient sound effects, action sound effects, and digital sound effects. Lastly, the students explore three types of music, including background music, theme songs, and mood music. Under the instructor's guidelines and demonstration, two students collaborate in groups to practice the *synchronized sound recording* for characters' conversations in MetaEcho. They also recorded the voices produced by the characters in the virtual set offered by MetaEcho, see Figure 2 (7c).

*Part 4 (10 minutes) Recording and Cinematography Collaboration*: this part details the cooperation between sound recordists and cinematographers. The shotgun mic's proximity to the performer directly affects the quality of the recorded sound in real-life filmmaking. Nevertheless, to preserve the visual appeal of the film, it is

necessary to ensure that the shotgun mic is not visible in the frame since the cinematographer makes adjustments to the angle and distance based on the plot of a play. Ensuring the excellent quality of recorded sound while preventing the unintentional visibility of the shotgun mic in the frame requires meticulous cooperation between the sound recordist and the cinematographer. Students replicate the camera's functioning in the MetaEcho system and engage in the practical exercise of recording dialogue situations while actors are walking. Figure 2 (6) shows the visual reference.

## 3.4 Implementation

To evaluate the efficacy, educational impact, and experimental performance of MetaEcho, we built the system utilizing the Unity platform, integrating Unity's multi-user networking capabilities to facilitate communication among users. We leverage Unity's networking module to track the location information and world views of virtual objects among all participants in real-time. Within the virtual environment of a multi-user setting, the teacher acts as the host and creates a virtual classroom that allows other students to join. The user locations and status information within this virtual room are shared and updated in real-time among all participants. Furthermore, we employed Blender to construct the three-dimensional models of MetaEcho scenes. The source code is available at https://anonymous.4open.science/r/MetaEcho-475B/README.md.

## 4 EXPERIMENTAL DESIGN

*Experimental Procedures.* Teaching modes were classified into two distinct categories. The first mode was using PowerPoint presentations with film case studies. These presentations were created to align with the course curriculum, which includes thorough written explanations and picture samples to illustrate the various knowledge and techniques. The duration of each class adhered precisely to the prescribed teaching timetable. The instructor employs vibrant language, accompanying visuals, and audio-visual aids in the PowerPoint presentation to elucidate the theoretical and practical prerequisites of simultaneous recording as outlined in the curriculum. The final component of the training involves collaboratively designing the programming to capture many events simultaneously. The instructor arranged the students into pairs based on the course schedule. Students were required to devise a creative and synchronised recording strategy for a specific filming situation. Every student group was allowed to provide an oral presentation showcasing their concept of SSR. Subsequently, the teacher offers comments. On the other hand, the second teaching

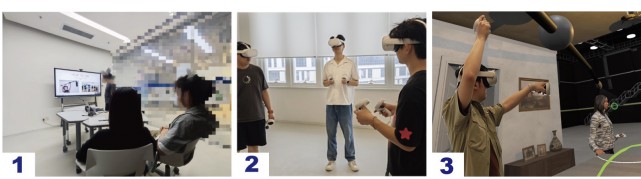

**Figure 3: Synchronized sound recording course: (1) Power-Point combined with video course; (2) - (3) MetaEcho.**

mode, also known as the MetaEcho condition, refers to learning in a virtual environment (Section 3.3). To evaluate the functionality and

teaching performance of MetaEcho compared to traditional classroom methods (PowerPoint, PPT) in the context of *synchronized sound recording* (SSR) instruction, as shown in Figures 3 (1) and (2)–(4). Both teaching scenarios utilise *synchronized sound recording* tutorials that have been reviewed by experts in the field of the *synchronized sound recording*. 12 students utilised MetaEcho to get instructions of the *synchronized sound recording* in a virtual studio, while another 12 students used PPT in a traditional classroom. The course content, training materials, and other learning resources are consistent between the two groups, with the only difference being the teaching medium. Before the start of the course, MetaEcho students get a 10-minute tutorial on using VR to ensure their familiarity with VR devices.

*Participants.* We recruited 24 participants from a university campus, consisting of 17 males and 7 females, within the age range from 21 to 30 years ($\bar{M}$ = 25.5, SD = 2.60). Half of these participants were randomly selected to partake in traditional teaching methods (PowerPoint (PPT) presentations with film case studies, i.e., the baseline). Another half received training in a virtual environment through MetaEcho's *synchronized sound recording*, which emphasized the practical application of skills in a virtual setting. All participants had either normal vision or vision corrected to normal standards. In the PPT group, all 12 participants had prior VR experience. In contrast, within the MetaEcho group, 10 of the 12 participants had experienced VR before. Participation was voluntary, and the university's institutional review board approved the experimental protocol. Each participant was compensated with a $5 upon completing the experiment.

*Tasks.* Next, all participants, regardless of their group, were told to do two evaluation tasks (Tests 1 and 2) to evaluate their learning effectiveness. To get consistent outcomes, individuals in both the PPT and MetaEcho groups were evaluated in the physical tasks known as Tests 1 and 2. The only difference was that individuals in both groups were exposed to different modes of teaching, either virtual reality or a traditional classroom presented using PowerPoint, before accomplishing Tests 1 and 2 of the physical condition.

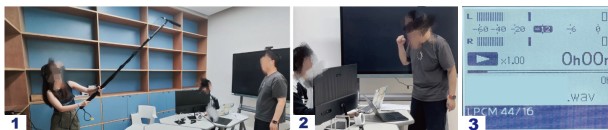

**Figure 4: (1): The scene of synchronized sound recording for a dialogue scene between two actors; (2): Boom operator proficiency test: a sample result; (3): A sample sound clip from the recorder screen display.**

*Test 1* aims to evaluate the boom operator proficiency test, and was designed to assess students' microphone pickup techniques in SSR. This test focuses on how boom operators accurately and efficiently handle the boom pole to optimally capture dialogue and sounds within a scene, ensuring the recording quality meets the high standards of film production[1, 39]. The learning outcomes of students after the MetaEcho or PPT teaching sessions are evaluated through expert scoring. In a real-world setting, each student has 10 minutes to use recording equipment (including a shotgun mic

and accessories, boom pole, recorder, and headphones) to record synchronized sound for a fixed shot of a two-person dialogue scene. The recorded sound must be of high quality, free of obvious technical errors, and the Shotgun mic must not mistakenly enter the frame, as shown in Figure 4 (1). Two authors, with rehearsal beforehand, participated as actors in all tests to maintain the consistency of the tests. Figure 4 (2) depicts a sample results. During the recording, the ideal RMS Level for human voices and dialogues should be within the range of -20dBFS to -12dBFS, as shown in Figure4 (3). The level is displayed on the recorder's screen through an LED indicator, allowing the boom operator to monitor the recording level to prevent distortion from levels that are too high or missing sound details from levels that are too low.

We invited three industry experts to score the participants' works. Experts 1 and 2 are from the film industry, specializing in synchronized sound recording, and have over five years of experience. Expert 3 is an acoustic expert from a music technology company who has participated in the design of several acoustic projects and has at least three years of industry experience.

We set a scoring protocol, employing a 7-point scale, as follows: The three experts conduct a blind review of the students' works based on sound quality (3 points), technical mistakes (3 points), and whether the shotgun mic mistakenly appears in the shot (1 point). The higher the score, the better the quality. Additionally, if the shotgun mic appears in the shot, one point is deducted. Finally, the average score from the three experts is computed, and an average score over 4 points (inclusive) passes the test; otherwise, it fails.

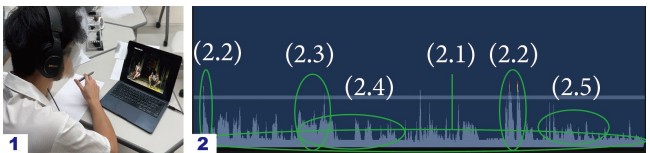

**Figure 5: (1): Synchronized sound recording listening test; (2): five results from the test video; (2.1): line interference (2.2): sound distortion; (2.3): accessory interference;(2.4): uneven volume; (2.5): off-axis actor dialogue.**

*Test 2* is a listening test of SSR evaluating the learning effectiveness of MetaEcho and PPT teaching methods. In the task, students identify various types of interference in synchronized sound recordings. Synchronized sound recording environments are subject to noise from multiple audio sources, as well as challenges such as line interference, sound distortion, accessory noise, uneven volume, and off-axis issues caused by actor movement. These challenges require operators to have comprehensive skills to minimize unnecessary interference and ensure recording clarity. During Test 2, participants watch a 44-second clip of raw film footage (Figure 5 (1)). They wear headphones to watch the video and identify five sound recording issues (Figure 5 (2)), including line interference, sound distortion, accessory interference, uneven volume, and off-axis actor dialogue. One point will be awarded for every issue identified. The summed score above three means passing the test; otherwise, it failed.

*After tests*: Upon finishing the two tasks, participants were instructed to complete a questionnaire about their learning experiences. The questionnaire on a 7-point Likert scale included 12 questions covering six different characteristics, with two questions dedicated to each topic, with 1 being the lowest and 7 representing the highest rating.

## 5 EVALUATION RESULTS

This section presents a quantitative analysis comparing student learning outcomes between MetaEcho and PPT in SSR for learning film production. Additionally, we report on user feedback and performance under both conditions.

### 5.1 Test 1: Boom Operator Proficiency Test

All 24 participants underwent an evaluation for boom operator proficiency, and three experts scored this evaluation.

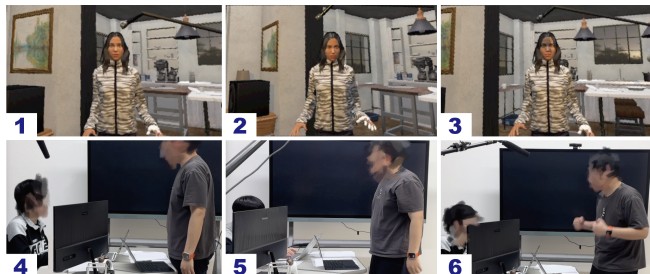

**Figure 6: Boom operator proficiency error examples: (1-4) Training in MetaEcho; (5-8) Testing in real environments.**

Figure 6 depicts examples of making mistakes in boom operator proficiency training within the MetaEcho environment (sub-figures 1-3) and compares them with examples of errors in real-world scenarios (sub-figures 4-6). We ran a Mann-Whitney U Test [25], showing the effect of teaching modes, with statistical significance between the two conditions ($U$ = 42.0, p <0.05). It is important to note that the tasks demonstrated a reasonable difficulty level because of the two actors in the scene. As such, boom operators are required to maintain the quality of SSR, simultaneously making no mistakes. Students who passed were assigned 1 point, and those who did not received 0 points. Figure 7 illustrates a sample of two participant's SSR. 10 out of 12 participants with MetaEcho succeeded in the test ($\bar{M}$ = 4.22, SD = 1.40), exceeding the threshold of 4 points (inclusive) on a 7-point scale. In contrast, only 5 out of 12 participants taught through traditional PowerPoint methods met the standard ($\bar{M}$ = 3.81, SD = 0.97), as shown in Figure 8 (1). This evidence of assessing boom operator proficiency implies that MetaEcho offers a more effective medium to deliver learning content than the PPT condition.

### 5.2 Test 2: SSR Listening Test

Figure 8 visualizes the results of Task 2. 9 out of the 12 participants with MetaEcho successfully passed the test to identify five types of SSR issues ($\bar{M}$ = 3.0, SD = 0.91). On the contrary, only 4 participants with the PPT condition passed the same test ($\bar{M}$ = 1.92, SD = 0.86). We ran a Mann-Whitney U Test [25], which shows statistical significance between the two conditions ($U$ = 42.0, p <0.05). The MetaEcho outperforms the traditional teaching model of PPT. The results imply that MetaEcho can promote students' ability to identify various problems in SSR. The key difference is due to

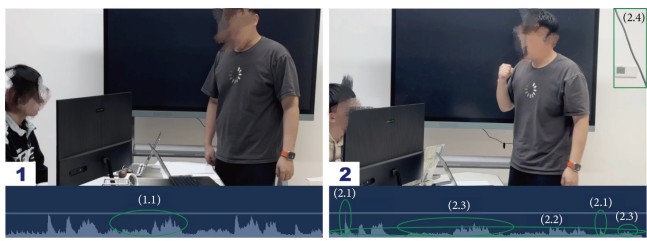

Figure 7: (1) MetaEcho: Operational mistakes: (1.1) Inconsistent loudness in actors' voices; (2) PowerPoint: Operational mistakes: (2.1) Sound effects are too loud or low; (2.2) RMS Level too low, resulting in unclear audio; (2.3) Inconsistent loudness in actors' voices; (2.4) Microphone wires visible in the shot.

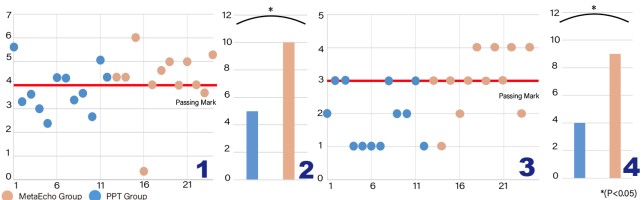

Figure 8: (1) Test 1 scores for boom operator proficiency, rated by the three experts; (2) Test 1 pass count; (3) Test 2 scores for SSR listening (X-axis: Participant IDs, and Y-axis: score); (4) Test 2 pass count (X-axis: two groups, and Y-axis: pass count).

MetaEcho's ability to visualize the auditory feedback, which reflects the shotgun mic's distance and direction in real time. Students with MetaEcho can remarkably establish concepts of SSR, including the source's orientation and distance. MetaEcho also shows great potential in enhancing students' understanding of SSR so they are more ready for complex acoustic environments.

### 5.3 Questionnaire Results

Figure 9 presents the values of the mean and standard deviation of the questionnaire of six dimensions (i.e., twelve questions), in which the blue and orange colour indicates the two conditions. Overall, MetaEcho outperforms the traditional method in all aspects. Furthermore, we conducted a Mann-Whitney U Test[25] and found statistical significance in all questions.

In the first dimension, for **Presence**, Q1"*I could feel like I was there to operate the boom*" ($U = 7.5$, p <0.001). Compared to the traditional PPT mode ($\bar{M} = 2.58$, SD = 1.51), participants using MetaEcho exhibited a stronger sense of operational realism ($\bar{M} = 6.00$, SD = 1.13). The teaching modules of MetaEcho offer an immersive experience in simulating SSR operations. Q2 "*I could feel the environment as if I were in the studio*" ($U = 9.0$, p <0.001). Participants using MetaEcho ($\bar{M} = 6.25$, SD = 0.62) exhibited a stronger immersive learning experience compared to those using PPT ($\bar{M} = 2.50$, SD = 1.73). For **collaboration**, Q3 "*I could communicate with teachers and students about the course content*" ($U = 6.5$, p <0.001) indicates the participants using MetaEcho could communicate better with teachers and other students ($\bar{M} = 6.67$, SD = 0.65) compared to those using PPT ($\bar{M} = 3.75$, SD = 1.82). Q4 "*I could work with teachers and*

*students to complete the task collaboratively*" reflects an obvious difference ($U = 12.5$, p <0.001). Compared to PPT-based teaching ($\bar{M}$ = 2.58, SD = 1.93), participants using MetaEcho exhibited stronger teamwork skills ($\bar{M} = 6.33$, SD = 0.89).

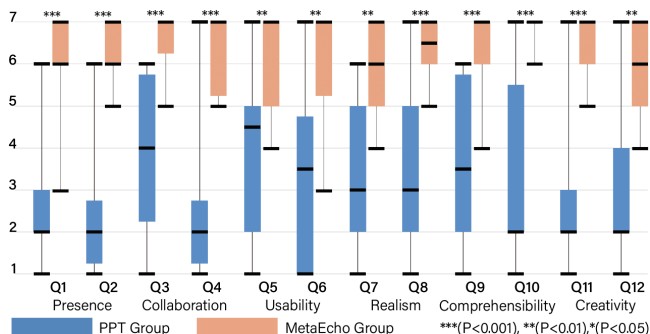

Figure 9: Participants' ratings on Presence, Collaboration, Usability, Realism, Comprehensibility and Creativity, ranging from 1 (Strongly Disagree) to 7 (Strongly Agree).

For **usability**, Q5 "*I found the synchronized sound recording instruction to be very functional*" ($U = 23.5$, p <0.01), participants using MetaEcho believed that the effectiveness of teaching SSR in the course ($\bar{M} = 6.17$, SD = 1.11) was superior to that of students using PPT ($\bar{M}$=3.83, SD=1.99). Q6"*I would choose the same mode of course delivery*" ($U = 18.0$, p <0.01). Compared to students using PPT ($\bar{M} = 3.17$, SD = 2.17) there was a higher level of acceptance for the course teaching method with MetaEcho ($\bar{M} = 6.17$, SD = 1.27). Regarding **Realism**, Q7 "*I think this course is sufficiently realistic to help my operation of synchronized sound recording equipment*" ($U = 17.0$, p <0.01), participants using PPT($\bar{M} = 3.25$, SD = 1.74) and MetaEcho ($\bar{M} = 5.92$, SD = 1.16) had differing levels of understanding when it came to operating SSR equipment. Q8 "*I think this course content is true to help understand synchronized sound recording*" ($U = 15.0$, p <0.01). Participants learning with MetaEcho felt they understood SSR better ($\bar{M} = 6.33$, SD = 0.78) compared to those using PPT ($\bar{M} = 3.42$, SD = 1.98). Next, **Comprehensibility**, Q9 "*I can understand how to conduct synchronous sound recording through descriptions of sound*" compared to PPT ($\bar{M} = 3.67$, SD = 1.92), students using MetaEcho ($\bar{M} = 6.33$, SD = 0.98) were able to better understand SSR through descriptions of sound, with a significant difference ($U = 14.0$, p <0.001). Q10 "*I can focus on audio sources in scenes to better understand types of audio sources*" ($U = 15.0$, p <0.001). MetaEcho participants ($\bar{M} = 6.83$, SD = 0.37) focused more on and understood sound sources better than PPT participants ($\bar{M} = 3.33$, SD = 2.13). Finally, the **Creativity**, Q11 "*I can focus on teaching scenarios to stimulate my creativity*" ($U = 11.0$, p <0.001), compared to the traditional PPT ($\bar{M} = 2.83$, SD = 1.77), participants using MetaEcho felt they could concentrate on the teaching scenario and were more creative ($\bar{M} = 6.50$, SD = 0.76). Q12 "*I can leverage variations in sound to design creative audio effects*" ($U = 17.5$, p <0.01). Compared to participants using PPT ($\bar{M} = 3.00$, SD = 1.83), MetaEcho participants believed they could better utilize changes in sound to design creative audio effects ($\bar{M} = 5.83$, SD = 1.14). This suggests MetaEcho has a greater advantage in inspiring creativity in sound among participants.

## 5.4 Expert Interviews

The evaluations from three experts on the MetaEcho system reveal its strengths and areas for improvement in SSR education. Expert 3 highlighted the effectiveness of MetaEcho in establishing the concept of SSR for beginners, especially its real-time feedback teaching mode, which can significantly reduce the learning curve for the novice. Expert 2 praised the immersive experience, believing it can help learners understand the role of a sound recordist and concentrate on the operation of the boom pole, thereby gaining a real filming experience. However, Expert 1 notes the MetaEcho system's limitations in offering in-depth teaching on advanced operational strategies for specific scenarios and microphone usage. Thus, Expert 1 proposed incorporating a diverse range of microphone models into the system. On the other hand, Expert 2 proposes customising the sound source material, the scenario, and the actor's movement track. By altering various settings and substituting sound sources that align with specific scenes, learners can gain a deeper comprehension of the attributes and manipulation techniques of sound in diverse settings, such as the interior of a spaceship or the exterior of a bustling city street. The simultaneous recording of sound sources in different environments will vary, and user customisation can enhance the system's versatility and suitability for educational purposes to the greatest extent.

## 6 CONCLUSION AND DISCUSSION

In this paper, we implemented and evaluated a metaverse-enabled learning and teaching tool for *synchronized sound recording* named MetaEcho. Our results show that learners with MetaEcho can effectively acquire the essential skills. Our discussion is as follows.

*Presence and visualization help understanding SSR.* In film education focused on recording techniques, VR presents a cutting-edge approach to address the intangible character of sound and alleviate the abstract concepts of SSR and hard-to-visualised components. Conventional instructional approaches, often focused on analysing sound via specific examples, provide challenges for learners in developing a clear understanding of auditory attributes, including closeness, intensity, and realism. Our method necessitates learners to comprehend sound attributes by engaging in repeated efforts and experimentation, resulting in a rapid learning progression [21].

*Realism encourages intuitive understanding of sound.* MetaEcho creates an immersive learning experience and realistic simulation of recording venues and equipment. This connects theoretical learning with hands-on exercises [11, 36]. As Test 1's results indicated, students using MetaEcho can directly interact with virtual recording equipment, making real-time recording adjustments. This direct interaction leads to a more intuitive understanding of sound characteristics and the practical application of recording techniques. Experts also agreed and were excited to include more realistic, sound material and environments in VR for SSR education.

*Real-time feedback improves performance.* The performance of the MetaEcho group in Test 2 was significantly better than that of the PPT group. It suggests that MetaEcho allows learners to receive immediate feedback on sound effects by adjusting the RMS level; moreover, altering the distance between the microphone and the actor produces varied auditory responses, which accelerates their

understanding of sound properties, as Expert 3 commented. Although MetaEcho demonstrates the potential to replace traditional PPT teaching methods for imparting SSR knowledge, the complexity of real SSR work scenarios and the significant differences in sound's acoustic properties across various physical environments necessitate a deeper exploration into areas such as acoustic physical simulation and interactive modes for future VR SSR teaching. Additionally, to better meet learners' needs and enhance teaching effectiveness, future research should explore more personalized learning pathways and teaching strategies that are adaptable to different learning styles.

*Require long-term practice for performance enhancement.* The results of Test 1 showed that VR demonstrated an advantage in increasing the pass rate of participants in short-term SSR training; however, while MetaEcho improved students' pass rates, it did not help participants reach higher scores. One reason is that, as Expert 1 suggested, MetaEcho focuses on basic practical operations and technical skills, while certain operations in SSR that require repeated practice, such as controlling the distance between the microphone and the actors of different loudness, are difficult to master through short-term VR learning.

*Film-making Education in the Metaverse.* MetaEcho serves as one example of metaverse applications, particularly in the context of film-making education, which broadens the knowledge of leveraging virtual worlds for art and culture, e.g., the latest work leverages VR for lighting training of film education [47]. We uniquely consider the deployment of SSR training, which works complementary to other works in the multimedia community. The findings provide evidence of MetaEcho's efficacy in increasing the learning experience, particularly in promoting presence, cooperation, usability, comprehensibility, and creativity. The coverage of film education in virtual worlds is expected to go beyond SSR and further encompass various roles in the training process, e.g., directors, cinematographers, and film artists. Moreover, film production involves many natural gestures among professionals. In contrast, our current VR relies on controllers, and thus, an opportunity exists to introduce natural gestures to facilitate user communication [2, 42, 48].

*Limitation and Future Work.* Although MetaEcho received participants' positive feedback, we identified a key limitation of the current system. The novice learners of MetaEcho encountered challenges in synchronising their movements with both the avatar and the camera. When the camera movement was rapid, constant adjustments in position were necessitated so that the users could align themselves with the avatar and the camera as they encountered them. As the users gain proficiency in operating the camera movement, unsuccessful synchronization is reduced. Our future plans include implementing an interface that allows learners to change the pace and intensity of camera movements during training, enhancing their ability to adapt to this teaching approach. Moreover, during our experiments, we observed that novice learners struggle to describe some film-specific terminologies accurately. Integrating visual clues in film teaching can help beginners better use film terminologies in their communication.

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
