# OpenReview forum: "Hearing the Moment with MetaEcho! From Physical to Virtual in Synchronized Sound Recording"
_acmmm.org/ACMMM/2024/Conference — MM2024 Poster_

### Official Review · Reviewer_VEr6 · 2024-05-07

**Rating:** 5
**Confidence:** 2

**Summary:**

This paper presents a virtual synchronized sound recording system.

**Strengths:**

1. The authors innovatively implemented a virtual reality environment to teach synchronized sound recording.
1. The proposed framework was evaluated thoroughly through two tests and expert interviews.
1. The code, questionnaire, and results are provided, increasing the credibility of this work.

**Limitations:**

1. The authors did not mention why they chose the kitchen and dining room from "It's Complicated" to replicate.

**Suitability:**

2

---

### Official Review · Reviewer_z9s5 · 2024-05-11

**Rating:** 5
**Confidence:** 3

**Summary:**

The paper introduces the design, implementation and evaluation of a VR-based teaching and learning system that allows users to learn synchronized sound recording more effectively without accessing the professional equipment. Results show that the system can lead to better learning outcomes and student experiences in comparison with conventional PowerPoint-based course content delivery.

**Strengths:**

The authors clearly demonstrated the limitations of sound recording education using conventional PowerPoint / discussion-based tools especially when access to expensive professional equipment is limited. The use of a 3D game engine and interactive virtual reality to simulate a sound recording environment and visualize the otherwise invisible sound-capturing metrics is novel. A user study with 24 participants was carried out. Both quantitative and qualitative metrics were used to measure the effectiveness of the system. The study also included expert opinions. The paper concludes with discussions of limitations and future work. The paper is well-written and easy to read.

**Limitations:**

The comparison to pure PowerPoint-based teaching does not seem to be "fair". I would imagine that some role-play using props would still be used in conventional teaching even when professional equipment is not available. It is not clear to me whether the users can walk to navigate or only use teleport/joystick function. I would like to see more discussions on the choice of different navigation methods. In my opinion, the proposed system is not something to replace PowerPoint/theory teaching but complements it. As the authors mentioned at the end of the paper, VR system has its own drawbacks. Your experiment includes 4 parts (page 4), can you summarize the impact (or the observed issues) of the new system in each of the 4 parts and perhaps how they are related to the tests you performed later on?

**Suitability:**

3

---

### Official Review · Reviewer_5WRP · 2024-05-25

**Rating:** 4
**Confidence:** 3

**Summary:**

the paper introduces the MetaEcho VR platform, which is designed for teaching Synchronized Sound Recording in film education. To this end, it provides realistic simulations of various recording equipment and offers the possibility for interaction between teacher and students. A subjective evaluation was conducted on 24 students, indicating its added value compared to traditional learning methods.

**Strengths:**

- The paper is well-written, has a clear structure, and is easy to follow. The contributions are relevant and timely
- Authors are addressing a very relevant topic within the boundaries of VR learning, with specific focus on a rather original use case.
- The methodology and evaluation procedure seem sound, and the obtained results are interesting and insightful

**Limitations:**

- After reading the manuscript, it is still not fully clear to me wat SSR concretely encompasses. I believe additional context and explanation on SSR is required in the introduction to better grasp its meaning and induced challenges in movie making. The same accounts for the concept of a shotgun mic. I believe not all readers are familiar with this equipment.
- Section 4: How were the questions of the questionnaire determined? Were these based on existing questionnaires? If not, how do the authors make sure not to bias participants to certain results?
- Section 6: In my opinion, it is a bit far fetched to describe the presented application as "metaverse-enabled". While being highly interesting and clearly useful, it remains a standalone VR application without a clear link to the actual Metaverse. Therefore, I would propose tuning down this claim. Furthermore, I think it would be an interesting and added value to the paper if the authors could shed light on what they believe to be the most important factors in design of applications for VR learning based on this use case.

EDITORIAL:
- Something seems off with the abbreviations. Typically these are written in full with the abbreviation between parentheses at first occurrence, e.g. PowerPoint (PPT), and only in short for every occurrence thereafter, e.g. PPT. This is currently not the case. Please check.
- Line 197: "therebyinfluencing" --> "thereby influencing" (add space)
- The numbering of figures is confusing, with both the main figure and its subfigures having numbers, e.g. Figure 2 (7a). I would propose reserving numbers for the main figures, and using letters for all subfigures.
- Figure 8: I believe it would be more informative to also represent these results as boxplots

**Suitability:**

3

---

### Official Review · Reviewer_pwU7 · 2024-05-26

**Rating:** 2
**Confidence:** 3

**Summary:**

The paper introduces MetaEcho, a VR platform designed to enhance the teaching and learning of synchronized sound recording in film education. Traditional methods of teaching SSR are limited by high costs, space constraints, and a focus on theoretical over practical experience. MetaEcho aims to address these issues by providing an immersive, interactive virtual studio that simulates real-world SSR scenarios.

**Strengths:**

1. MetaEcho represents a pioneering effort to use VR for teaching synchronized sound recording, an area traditionally underserved by immersive technologies. This innovation bridges the gap between theoretical knowledge and practical skills in film sound recording education.

2. The paper is well-organized and clearly written, with detailed descriptions of the system design, experimental procedures, and evaluation results. The inclusion of figures and diagrams enhances understanding.

3. The evaluation methodology is rigorous, involving a controlled experiment with 24 students and a detailed analysis of learning outcomes.

**Limitations:**

1. While MetaEcho effectively teaches basic SSR skills, it does not address advanced operational strategies or specific microphone usage scenarios in depth.

2. Novice users experienced challenges in synchronizing their movements with the avatar and camera, particularly during rapid camera movements. This indicates a need for improved user interfaces or training modules to better prepare students for complex VR interactions.

3. The author compares MetaEcho primarily with traditional PowerPoint-based teaching. It would be beneficial to include comparisons with other emerging educational technologies, such as augmented AR or other VR platforms, to contextualize MetaEcho’s effectiveness within a broader spectrum of tools. The current experiments are not enough to explain the advantages of MetaEcho in teaching.

4. The conclusion is not strong enough, and it does not present novel or surprising findings that differentiate it from other VR teaching systems.

**Suitability:**

2

---

### Meta-Review · Area_Chair_KhCE · 2024-07-02

**Recommendation:** Accept (Poster)
**Confidence:** 4

**Metareview:**

The paper introduces MetaEcho, a VR platform designed to enhance the teaching and learning of synchronized sound recording (SSR) in film education. High costs, space constraints, and a focus on theory over practical experience limit traditional methods of teaching SSR. MetaEcho aims to address these issues by providing an immersive, interactive virtual studio that simulates real-world SSR scenarios.

The paper has received mixed feedback from the reviewers and is at a borderline decision as they identified various strengths and weaknesses.

Reviewer pwU7: Highlights the effort to use VR for SSR but notes limitations in addressing advanced operational strategies and the narrow comparison with traditional PowerPoint-based teaching.
Reviewer 5WRP: Highlights the clear structure and contributions of the paper but finds the range of scenarios limited and the comparison with PPT-based training insufficient to prove its strength.
Reviewer z9s5: Values the novel use of VR in professional training and the comprehensive evaluation but notes that the comparison with PowerPoint-based teaching is not entirely fair, and the navigation methods are not well-discussed.
Reviewer VEr6: Supports the innovative implementation and thorough evaluation but suggests additional experiments to strengthen the conclusions.

The paper is well-written, and the approach is innovative. However, improvements should be made to address the limitations highlighted by the reviewers. I am not sure the authors have enough time to address these concerns to produce a publication ready paper by the conference deadline.

Several common limitations have been identified:
1. Scope of SSR: The paper does not provide sufficient context and explanation about what SSR encompasses, which could hinder understanding for readers unfamiliar with the concept.
2. User Experience: Novice users faced challenges in synchronizing their movements with the avatar and camera, particularly during rapid movements, indicating a need for improved user interfaces or training modules.
3. Comparative Analysis: The comparison is primarily with traditional PowerPoint-based teaching. Including comparisons with other emerging educational technologies, such as AR or other VR platforms, would provide a broader context of MetaEcho’s effectiveness.
4. Experimental Scope: The experiments are limited in scope, and the conclusion does not present novel or surprising findings that differentiate MetaEcho from other VR teaching systems.
5. Novelty of findings: The limited value of the findings and conclusions has also been highlighted.

Based on the reviews, the paper is recommended for weak rejection. The innovative approach and comprehensive evaluations are significant strengths, while the concerns regarding experimental robustness and comparative analysis warrant attention. I would strongly encourage authors to present this research as a poster.